# Capillary Zone Electrophoresis in Tandem with Flow Cytometry in Viability Study of Various ATCC Bacterial Strains under Antibiotic Treatment

**DOI:** 10.3390/ijerph19031833

**Published:** 2022-02-06

**Authors:** Wojciech Kupczyk, Ewelina Maślak, Viorica Railean-Plugaru, Paweł Pomastowski, Marek Jackowski, Bogusław Buszewski

**Affiliations:** 1Department of General, Gastroenterological, and Oncological Surgery Collegium Medicum, Nicolaus Copernicus University, Gagarina 7, 87-100 Toruń, Poland; kupczykwojciech@o2.pl (W.K.); jackowscy@hotmail.com (M.J.); 2Centre for Modern Interdisciplinary Technologies, Nicolaus Copernicus University in Toruń, Wileńska 4 Str., 87-100 Toruń, Poland; e_maslak@doktorant.umk.pl (E.M.); viorica.railean@umk.pl (V.R.-P.); pomastowski.pawel@gmail.com (P.P.); 3Department of Environmental Chemistry and Bioanalytics, Faculty of Chemistry, Nicolaus Copernicus University in Toruń, Gagarina 7 Str., 87-100 Toruń, Poland

**Keywords:** capillary zone electrophoresis, flow cytometry, bacterial viability, antibiotics, pathogenic strain

## Abstract

The aim of this study was to develop an innovative method of examining bacterial survival using capillary zone electrophoresis (CZE) and flow cytometry (FC) as a reference method. For this purpose, standard strains of bacteria from the ATCC collection were used: *Enterococcus faecalis* ATCC 14506, *Staphylococcus aureus* ATCC 11632, *Klebsiella pneumoniae* ATCC 10031, *Pseudomonas aeruginosa* ATCC 27853, and *Escherichia coli* ATCC 25922, as well as seven antibiotics with different antimicrobial mechanisms of action. The ratio of live and dead cells in the tested sample in CZE measurements were calculated using our algorithm that takes into account the detection time. Results showed a high agreement between CZE and FC in the assessment of the percentage of live cells exposed to the stress factor in both antibiotic susceptibility and time-dependent assays. The applied measuring system to assess the effectiveness of antibiotic therapy in in vitro conditions is a method with great potential, and the data obtained with the use of CZE mostly correspond to the expected drug sensitivity according to EUCAST and CLSI guidelines.

## 1. Introduction

The bacterial cell is forced to constantly adapt to new changing environmental conditions, such as temperature, pressure, oxygen and carbon dioxide levels, or the presence of water, nutrients or toxic substances. These parameters influence its metabolic activity and change its physiological state [1]. From the point of view of many microbiological assays, it is particularly important to monitor the physiological state of bacteria. The crucial element in laboratory diagnosis, besides identifying pathogens, is the assessment of their viability and antimicrobial susceptibility testing mostly determine by phenotypic (e.g., minimum inhibitory concentration (MIC) or genotypic approaches, which is essential to understanding the pathogen’s response to therapy [2]. The viability is identified as the ability of a cell to grow and reproduce itself under a set of defined environmental conditions. Viable cells are characterized by the presence and functioning of a range of structural, physiological, metabolic, and genetic properties [3].

Generally, it is possible to distinguish four groups of cells characterized by a different physiological state. The most metabolically active group was defined as alive bacteria which are able to grow on laboratory media. Under certain conditions, living bacteria can go into a reversible state of anabiosis, in which they remain viable but non-culturable (VBNC). These cells have lost the ability to grow on the routine bacteriological media on which they would normally grow and develop into colonies [4,5]. The next group are dead bacteria that do not exhibit metabolic activity. On the basis of the coherence of their cell membrane, they can be divided into dead cells with an intact cytoplasmic membrane and dead cells after lysis of the cytoplasmic membrane [1].

There are two methodologies for determining the viability of a microbial population: culture-dependent and culture-independent. The first one mainly uses the ability of cells to multiply on the surface of solid media under the conditions provided for growth. For this reason, their use is associated with a long waiting time for the appearance of colonies, and this varies from days to weeks depending on the bacteria. For most bacteria, it is required to cultivate them for 24–48 h. In this study, laser desorption ionization mass spectrometry was also applied as part of antimicrobial susceptibility testing [6]. In conventional techniques, viability depends exclusively on the ability of the cell to reproduce with a visual inspection of the colony formed from the original single cell. It makes these techniques non-adaptable for damaged or stressed, non-reproducing cells (VBNC) which can result in false negatives [2,7]. A range of culture-independent technologies have been developed to differentiate between live and dead bacteria quickly and easily. They rely, for example, on the detection of metabolic activity, transcriptional capacity, and membrane integrity [6]. Flow cytometry (FC) is a popular diagnostic technique that enables qualitative (e.g., viability) but also quantitative analyses of bacterial cells. This technique enables the simultaneous multi-parameter analysis of chemical and/or physical properties (size, shape, and the macromolecular content) of over 50,000 particles per second [8,9]. The bacteria cell surface or components must first be labelled. In fluorescence flow cytometry, which is the standard used, cells are labelled with one or more fluorescent dyes. Cells are aligned to pass individually through a laser beam, which excites fluorescent particles. This results in the emission of light at different wavelengths. The amount and type of fluorescence indicates the percentage of different cell types or cell components in the sample [1]. The type of dye used determines the analysed parameters and the sensitivity of detection. For the differentiation of living and dead cells, dyes such as propidium iodide (PI) or SYTO 9 are used. The signals from the dyes can also be measured using fluorescence microscopy or fluorescence-based microplate readers [10]. Recently, mass cytometry become a complimentary technique to FC for the identification and analysis of bacteria. In this technique, instead of fluorescent dyes like in standard FC, cells are labelled with stable isotopes of transition metals (mainly lanthanides) [11]. Cells thus isotope-labelled are detected by an inductively coupled plasma mass spectrometry, which because of high resolutions enables simultaneous analysis of even more than 30 parameters [12].

In this study, we investigated the influence of various antibiotics on ATCC bacterial strains. For each strain, antibiotics were selected separately according to EUCAST and CLSI guidelines. Moreover, the bacterial viability evaluation has been performed using a complimentary approach; initially, optical density (OD) measurements were carried out, followed by capillary zone electrophoresis (CZE) analysis, and confirmed by a standard flow cytometry technique. The OD at a wavelength of 600 nm is the most common method for estimating the number of cells in a liquid suspension. Although this technique does not allow for the distinguishing of whether bacterial cells are alive or dead, it can be helpful during the establishing of a calibration method [13]. Electrophoretic techniques enable the movement of microorganisms in an electric field, which allows them to be used for separation purposes [14]. In previous studies, Buszewski et al., with the use of CZE, successfully separated bacterial species qualitatively, achieving better and better selectivity against *Staphylococcus aureus* [15] and *Escherichia coli* [16]. To our knowledge, this study is the first to successfully use CZE to quantify the viability of bacteria subjected to antibiotic stress.

## 2. Materials and Methods

### 2.1. Preparation of Biological Material

In the present study, different bacterial strains from the American Type Culture Collection (ATCC) were used: *Enterococcus faecalis* ATCC 14506, *Staphylococcus aureus* ATCC 11632, *Klebsiella pneumoniae* ATCC 10031, *Pseudomonas aeruginosa* ATCC 27853, and *Escherichia coli* ATCC 25922. Initially, all bacterial strains were aerobically revitalized using a solid culture medium—Tryptic Soy Agar (Sigma Aldrich, Germany)—at 37 °C for 24 h. The bacterial colonies were transferred to the liquid Mueller Hinton Broth medium (Sigma Aldrich, Germany) and an inoculum = 1 × 10^6^ CFU/mL (colony forming unit per millilitre) was prepared.

### 2.2. Preparation of Antibiotics

For this step, seven antibiotics have been chosen: amoxicillin, amoxicillin with clavulanic acid, cefotaxime, clindamycin, ciprofloxacin, vancomycin, metronidazole (all purchased from Sigma Aldrich, Steinhelm, Germany). The antibiotics were dissolved in deionized water obtained from the Milli-Q apparatus (Millipore Intertech, MA, USA). The stock solutions were prepared in such a way as to obtain the final concentration of the system as follows: amoxicillin—0.1 mg/L, 2 mg/L; amoxicillin with clavulanic acid—0.1 mg/L, 1 mg/L, 2 mg/L, 4 mg/L, 8 mg/L; cefotaxime—0.5 mg/L, 1 mg/L, 2 mg/L, 4 mg/L, 8 mg/L; clindamycin—0.1 mg/L; ciprofloxacin—0.1 mg/L, 1 mg/L, 2 mg/L, 4 mg/L, 8 mg/L; vancomycin—0.5 mg/L, 1 mg/L, 2 mg/L, 4 mg/L, 8 mg/L; metronidazole—0.1 mg/L, 0.5 mg/L, 3 mg/L.

The selection of antibiotics and their concentrations for the tested strains was based on the EUCAST and CLSI guidelines. For each bacterium, a compound was selected as the lead antibiotic that is described in the above-mentioned guidelines—it had specific minimal inhibitory concentration values on the basis of which its sensitivity or resistance could be determined.

### 2.3. Antibiotic Susceptibility Assays (Static Assays)

For each bacterial strain, three types of cells were tested: live—control (without antibiotic), dead (inactivated with 70% ethanol), and treated with antibiotics. In the antibiotic susceptibility assays, the antibiotic effect has been studied over a wide range of concentrations after 24 h of incubation. For *P. aeruginosa* it was cefotaxime, for *E. faecalis* it was vancomycin, for *S. aureus* it was ciprofloxacin, and for *E. coli* and *K. pneumoniae* it was amoxicillin with clavulanic acid. For the analysis, bacterial cells were mixed (1:1) with selected antibiotic drugs at already prepared concentrations (indicated in Section 2.2) and incubated for 24 h, according to the CLSI (Clinical Laboratory Standards Institute) and EUCAST (European Committee for Antimicrobial Susceptibility Testing) guidelines. After this time, the samples have been subjected to the following analysis: OD measurements, flow cytometry and capillary zone electrophoresis for the viability investigation. In all cases, three replicates were performed.

### 2.4. Time-Dependent Assays (Dynamic Assays)

At this step, for each bacterial strain two types of cells were tested: live—control (without antibiotic), and treated with antibiotics. For each strain, the leading antibiotic was applied (the same as in the antibiotic susceptibility assays), whose effects have been studied after different periods of incubation time. The dynamic study was performed by determining the bacterial growth after 1, 12, and 24 h of incubation with the selected antibiotics at a constant concentration of 8 mg/L. After this time, the samples were subjected to the Optical Density Measurements (OD) measurement, flow cytometry and capillary zone electrophoresis for the viability investigation. In case of all analysis, three repetitions have been taken in consideration and presented as a form of standard deviation. The schematic course of the experiment is shown in Figure 1.

### 2.5. Optical Density Measurements (OD)

All prepared samples, as is described in Section 2.3 and Section 2.4, were measured at the wavelength at λ = 600 nm. For this purpose, a UV-Vis spectroscope with a 96-well plate reader (Multiskan™ FC Microplate Photometer) was used. During the measurements, the effect of its own background absorbance by the pure medium was taken into account. Three replicates of each sample were made.

### 2.6. Capillary Zone Electrophoresis Measurements

For this purpose, the untreated and treated cell with antibiotics have been prepared as is described in Section 2.3 and Section 2.4. After 24 h of incubation in case of each experiment (antibiotic susceptibility and time-dependent assays), bacterial cells were washed twice with deionized water. To modify the surface of the microbes, the purified bacterial pellet was suspended in 0.005 M Ca(NO3)2 and incubated for one hour at room temperature. After this time, the precipitate was centrifuged (4 °C, 4500 rpm, *t* = 15 min), and the supernatant was removed and successively washed with deionized water. Purified from excess calcium ions, the bacterial pellet was suspended in TB buffer (TRIS CTRIS = 4.5 mM, boric acid CB = 50 mM, pH 7.98–8.3) and subjected to electroanalysis by means of Capillary Zone Electrophoresis (CZE). For this purpose, an HP3DCE apparatus with a diode array detector (DAD) and quartz capillaries (Id = 75 μm; Ltot = 33.5 cm; Leff = 25 cm; Composite Metal Services, Shipley, UK) were used. The inner surfaces of the quartz capillary were activated by rinsing them sequentially: 1.0 M NaOH, 0.1 M NaOH, 0.1 M HCl, deionized water, BGE and outlet buffer for 5 min each. The electrophoretic analysis was performed in a non-linear system using the following buffers: outlet—TB, and inlet—TB-HCl (CTRIS = 4.5 mM, CB = 50 mM, CHCl = 4.4 pH 7.35–7.49). The analyzes were carried out at a maximum work intensity of Imax = 100 μA, constant voltage U = 15 kV and a temperature of 23 °C. The wavelength λ = 214 was used to monitor the electromigration of bacterial aggregates. Bacterial cells were introduced into the capillary by applying a vacuum of 50 mbar for 25 s. The capillary was conditioned by rinsing the system with 1.0 M NaOH and deionized water for 2 min prior to each electroanalysis. The electroosmotic flow was determined using acetone.

Three replicates of each sample were made, and the obtained signals were analyzed and integrated using the ChemStation Off-line software package. To determine viability, the retention times for live cells (no antibiotic addition) and dead cells (live cells inactivated with 70% ethanol) were first determined. The retention time was calculated for each spectrum—the average of all signals present in the electropherogram. The mean values with the standard deviation for the tested bacterial strains are presented in Figure 2. In each case, the dead cells reached the detector in less time than the living cells. Then, measurements of bacteria incubated with antibiotic solutions were performed, and the retention time of cells treated with antibiotics was determined. Its value was within the time limits of live and dead cells for a given strain and depended on the used antibiotic and its concentration (see Figure 2). Based on these three retention times, the percentage of living and dead cells was calculated according to the proposed Equations (1) and (2). Where: ***X***[%]—percentage of live cells, ***Y***[%]—percentage of dead cells, *t_A_*—retention time of dead cells, *t_B_*—retention time of live cells, *t_i_*—retention time of cells treated with an antibiotic. Surface area of each signal recorded in electropherograms has been measured and the sum of the all of them was calculated.
(1)$$X\left[\%\right]=\frac{t_{i}-t_{A}}{t_{B}-t_{A}}\cdot 100\%$$
(2)Y[%]=100%−X[%]

### 2.7. Flow Cytometric Measurements

Flow cytometry was carried out using a MACSQuant VYB (Miltenyi Biotec, Clopper Road, MD, USA) Cytometer. To analyse the viability of bacterial cells, 1 µL of propidium iodide (Sigma Aldrich, Poznań Poland) solution at a 100 µg/mL concentration was added to each sample. Samples were shaken and incubated in the dark for 30 min at room temperature. Bacterial cells were analysed using a blue laser (488 nm) at channel pairs of 655–730 nm and 585/40 nm, in order to differentiate the live and dead cells. All measurements were carried out in triplicate. The obtained data were subjected to a gating procedure using the MACSQuantify program, and then they were processed using standard spreadsheet software (Excel).

### 2.8. Statistical Analyses

All data were processed using a complimentary approach. The Microsoft excel 2016 was used for both processing of raw data as well as for performing the Pearson correlation coefficient test. Additionally, the correlation analysis of the obtained data was used based on the Student’s *t*-test for *p* = 0.05. Two-dimensional scatter plots with the regression lines was used to determine the relationship between the variables OD and FC using STATISTICAL Release 7 software.

## 3. Results

Two different approaches were considered to achieve the aim of our study, which was to determine the suitability of capillary zone electrophoresis in assessing the in vitro efficacy of antibiotics against five bacterial strains (*Enterococcus faecalis* ATCC 14506, *Staphylococcus aureus* ATCC 11632, *Klebsiella pneumoniae* ATCC 10031, *Pseudomonas aeruginosa* ATCC 27853, and *Escherichia coli* ATCC 25922). In antibiotic susceptibility assays, seven antibiotics belonging to different classes which present different mechanisms of action on bacteria were used: penicillins (amoxicillin, amoxicillin with clavulanic acid), cephalosporins (cefotaxime), fluoroquinolones (ciprofloxacin), glycopeptides (vancomycin), lincosamides (clindamycin), and miscellaneous (metronidazole). Antibiotics and their concentrations were selected individually for each strain according to EUCAST (European Committee for Antmicrobial Susceptibility Testing) and CLSI guidelines. In the time-dependent assays for selected antibiotics, the effect of incubation time on the survival of the tested strains was determined.

### 3.1. Optical Density

Optical density measurements allowed for the initial estimate of possible changes after the addition of antibiotics. In antibiotic susceptibility assays, compared to the control samples, a decrease in the OD value was observed after the addition of the antibiotic in the case of all tested samples (Figure 3). The mean OD value for the control samples was 0.95, while for the samples after the addition of the antibiotic it was 0.28. Metronidazole was the only antibiotic used for all tested strains. The most significant decrease in the optical density value was observed in the case of *K. pneumoniae* (decrease of about 71%), and the smallest was in the case of *P. aeruginosa* (only 13%). In the case of the remaining strains, the OD values decreased by about half. The use of metronidazole in the concentration limit of 0.1–3 mg/L did not significantly affect the measured values in the case of *S. aureus*. Clindamycin was used for all strains except *E. coli*. In the case of *K. pneumoniae*, its use significantly lowered the OD value (decrease by 60%). For the remaining strains, the decrease was approximately half. Cefotaxime applied against *E. coli* caused the OD value to drop by approximately 40%. A decrease of about half was observed for the *E. faecalis* strain. The use of cefotaxime significantly lowered the measured value for *P. aeruginosa* (reduction by over 93%). Moreover, in this case, the use of its different concentrations did not affect the change in optical density. The use of vancomycin against *E. coli* slightly changed the OD value. In the case of *K. pneumoniae*, a better effect was achieved—a decrease of approx. 65%. In the case of *E. faecalis*, the use of vancomycin caused the most significant reduction in value (over 86%), which remained stable despite the use of higher concentrations of antibiotics. Amoxicillin was used in the analysis of the three strains. Compared to the control sample, in the case of *P. aeruginosa*, its use resulted in the most significant difference in OD values. Slightly higher OD values were obtained for the *S. aureus* strain and the highest was for *E. coli*. For *P. aeruginosa*, the decrease in the OD value was slight, for *E. faecalis* it halved, and the use of this antibiotic against *Enterobacteriaceae* was the most influential. Moreover, the use of amoxicillin with the addition of clavulanic acid in a different concentration range (0.1–8 mg/L) did not significantly decrease the OD value for *E. coli* and *K. pneumoniae*. The effect of ciprofloxacin was investigated for only two strains—*K. pneumoniae* and *S. aureus*. The use of a concentration of 0.1 mg/L significantly lowered the OD value in the case of *K. pneumoniae* (decrease by 84%). In the case of *S. aureus*, only the increase of the applied concentration to more than 1 mg/L allowed for the obtaining of a similar effect. In the time-dependent assays, the optical density value slightly decreased after one hour of incubation in the case of gram-negative strains (*E. coli*, *K. pneumoniae*, *P. aeruginosa*). In the case of *S. aureus*, a slight decrease was observed after 12 h. After this time, the OD values started to increase again. In the case of *E. faecalis*, the measured value increased after 12 h and decreased after 24 h.

### 3.2. Capillary Zone Electrophoresis Assay

In antibiotic susceptibility assays, the addition of the antibiotic in each case lowered the percentage of live bacterial cells, increasing the rate of dead cells. The percentage of live and dead bacteria is presented on the example of the *E. feacalis* strain in Figure 4A. (see Figure 4A). Metronidazole at a concentration of 0.1 mg/L used for all strains had the strongest effect on the viability of *S. aureus*—the viability of the cells was reduced by half. In the case of the remaining strains, the percentage of dead cells ranged from 25% to 35%. For *S. aureus*, its effect has also been studied at higher concentrations. With the use of 0.5 mg/L, the percentage of viable cells was the highest (about 79%), and with the use of 0.1 and 3 mg/L comparable. In the case of clindamycin at a concentration of 0.1 mg/L, the lowest antibacterial effect was observed for gram-negative strains of *K. pneumoniae* and *P. aeruginosa*—the percentage of dead cells was 15% and 18%, respectively. In turn, cefotaxime was the least effective against *E. coli*—81% of the bacterial cells survived. A slightly better result was obtained for the *E. faecalis* strain. In addition, this result was slightly worsened by using a higher concentration of the antibiotic. Cefotaxime showed the best efficacy against *P. aeruginosa*, which increased in proportion to the concentration. Vancomycin was the least effective for *K. pneumoniae*—over 81% of bacterial cells remained alive. A higher effect was obtained for the *E. coli* strain. In the case of *E. faecalis*, the use of the lowest concentration of vancomycin (0.1 mg/L) allowed the reduction of viable cells to 28%. Additionally, the increase in concentration slightly decreased this value. The low amoxicillin concentration was very favorable for *S. aureus*—it lowered the cell viability to 10%. In the case of *E. coli*, the viability was reduced the least—to 63%. In contrast, in the case of *P. aeruginosa*, the use of amoxicillin at a concentration of 0.1 mg/L and 2 mg/L caused the percentage of viable cells to increase with increasing drug concentration. The 0.1 mg/L of amoxicillin with clavulanic acid had the lowest effectiveness against *K. pneumoniae*—84% of cells remained alive, and the highest in the case of *P. aeruginosa*—54% were alive. It showed a similar effect on *E. coli* and *E. faecalis*—about 67% of the cells remained alive. The use of higher concentrations against *Enterobacteriaceae* significantly reduces cell viability. Ciprofloxacin reduced *K. pneumoniae* cell viability to 63%. And in the case of *S. aureus*, the use of a higher concentration linearly reduces the percentage of viable cells.

In the time-dependent assays, the incubation time of bacterial cells with a constant concentration of antibiotics showed an effect on the viability of bacterial cells. The percentage of live and dead bacteria is presented on the example of the *E. feacalis* strain in Figure 4B. Amoxicillin with clavulanic acid showed a similar effect for both strains of enterobacteria (*E. coli*, *K. pneumoniae*). After one hour, the percentage of viable cells was approximately 40%, and after 12 h, about 10%. Further incubation slightly reduced the viability. In the case of the remaining strains, the most significant decrease in viability was also observed after one hour of incubation; however, the extension of the incubation time did not have a significant effect on the viability of the bacteria. Cefotaxime reduced the viability of *P. auregionosa* by half, ciprofloxacin lowered the viability of *S. aureus* to about 33%, and vancomycin applied to *E. faecalis* decreased this value to 14%. The results for all strains expressed as percentage of viable cells are presented in the Appendix A.

Additionally, for each strain, eight bacterial suspensions (without the addition of antibiotics) were prepared, for which the OD values were measured. Figure 4C shows the total area of all signals of *E. faecalis* that was obtained from the electrophoretograms. These values were compared to investigate the relationships between CZE/FC; calibration curves were plotted. An example of a calibration curve is presented in Figure 4D. In case of all investigated bacterial strains the same approach has been applied. A very high correlation was obtained between the absorbance values (OD) and the peak area on the electropherogram—R2 values > 0.99 for all tested strains. The highest match was obtained for the *E. coli* strain, and the lowest for *S. aureus*. These results indicate that the optical density is related to the peak area of the electropherogram.

### 3.3. Flow Cytometry Measurement (FC)

FC analysis enables the control of the inhibition of the growth of microorganisms by the simultaneous determination of the number of living and dead cells. In antibiotic susceptibility assays, the addition of the antibiotic lowered the percentage of live bacterial cells, and increased the rate of dead cells. The obtained values were in most cases similar to the values obtained with the use of CZE. The percentage of live and dead bacteria obtained by CZE is presented on the example of the *E. feacalis* strain in Figure 5A. The use of metronidazole reduced the viability of *S. aureus* cells the most—the percentage of viable cells was 47%, while dead cells increased to the 53%. In the case of *P. aeruginosa* viability was up to 65%, *E. faecalis* and *K. pneumonaie* up to 75% and *E. coli* up to 89%. Clindamycin turned out to be the least effective in the case of *K. pneumonia*, and *P. aeruginosa*—the percentage of dead cells was below 13% and 21%, respectively. Its use in *E. faecalis* reduced the viability to almost 70% (30% of cells was dead) and *S. aureus* to almost 47% (dead 53%). *E. coli* demonstrated the greatest resistance to the action of cefotaxime—bacterial viability after its application was approximately 96%. A better effect was achieved with *E. faecalis*, with a viability of 75%; however, using a higher concentration of the antibiotic slightly increased the percentage of viable cells, whereas the rate of dead cells lowered. The best effectiveness of cefotaxime was shown with *P. aeruginosa*. Its lowest concentration reduced the viability of this bacteria by half, and increasing the concentration to 8 mg/L reduced the viability to about 17%. Vancomycin reduced the viability of *E. faecalis* to a greater extent than that of the gram-negative strains. The use of the lowest concentration against *E. faecalis* lowers the viability to 38%, and its increase reduces this value. The use of amoxicillin had the most significant impact on the viability of *S. aureus*—the use of a concentration of 0.1 mg/L reduced the percentage of viable cells to 9%, while the percentage of dead increased to 91%. For *P. aeruginosa*, the same concentration reduced the viability to 31%, but a concentration of 2 mg/L increased this value to 71%. The percentage of live *E. coli* cells for the lowest concentration was almost 83%. Amoxicillin with clavulanic acid showed the lowest effectiveness against enterobacteria—the use of a concentration of 0.1 mg/L reduced the viability to 88% in the case of *E. coli* and 84% in the case of *K. pneumoniae*. A significant decrease in viability was observed at a 1 mg/L concentration for *E. coli* and 4 mg/L for *K. pneumoniae*. Ciprofloxacin lowered the viability of *K. pneumoniae* to 75% and *S. aureus* to 37%. Higher concentrations allowed for the reduction in the viability of *S. aureus* cells significantly—at the concentration of 4 mg/L, the percentage of living cells was approximately 3%.

In the time-dependent assays, an hour-long incubation significantly reduced the viability of bacterial cells. Figure 5B presents the percentage values for *E. faecalis*. The exception was the enterobacteria incubated with amoxicillin with clavulanic acid. After one hour, the viability of *K. pneumoniae* and *E. coli* decreased to 65% and 54% respectively. After 12 h, the viability had significantly reduced to about 1% for *K. pneumoniae* and 4% for *E. coli*. In the case of the remaining strains, the most significant decrease in viability was observed after 1 h of incubation with antibiotics. The viability of *P. aeruginosa* cells after incubation with cefotaxime was 73%, *S. aureus* with ciprofloxacin 17%, and *E. faecalis* with vancomycin 11%. Further incubation slightly decreased this value. Figure 5C shows the viability of some of the cells recorded by Flow Cytometry technique. The results for all strains expressed as percentage of viable cells are presented in the Appendix A.

### 3.4. Capillary Zone Electrophoresis and Flow Cytometry Data Correlation

The viability of bacterial cells was determined using two methods: capillary zone electrophoresis (research method) and flow cytometry (reference method). Curves, showing the dependence of the percentage of obtained viable bacterial cells using both ways, were plotted from the results obtained in the antibiotic susceptibility assays (Figure 6). The results obtained with CZE were much more often underestimated in relation to FC. The exception is the *S. aureus* strain for which the reverse was observed. The control samples showed a discrepancy ranging from 0.23% for *S. aureus* to 1.42% for *P. aeruginosa*. The exception was the control sample of *E. coli*, for which the difference in measurements was over 8%. In the case of the control samples, larger deviations were observed. The smallest discrepancy in the results was obtained in the case of *P. aeruginosa*—on average the results differed by 3.49%. In the case of *K. pneumoniae* and *E. faecalis*, the mean difference was similar and amounted to 4.28% and 4.31%, respectively. For *S. aureus* it was 6.46% and for *E. coli* 14.55%. In the case of *P. aeruginosa*, the difference in the measurements of bacterial viability was the lowest for metronidazole (difference 0.23%). The use of cefotaxime (especially in high concentrations) gave the least consistent results. Altogether, in 9 out of 10 measurements, the differences between the results did not exceed 5%. In the case of *E. faecalis*, the smallest difference was also observed for cells incubated with metronidazole (0.25%). In five measurements, the difference exceeded 5%, in each case it was related to vancomycin (the biggest difference was 13%). *S. aureus* showed the slightest discrepancy in the results after the application of amoxicillin and the highest concentration of ciprofloxacin. However, for the lower concentrations of ciprofloxacin, a significant difference was observed in the measurements—for the concentration of 1 mg/L was over 20%. In total, in the four measures, the values exceeded 5%. The viability measurements for *K. pneumoniae* were most similar for the lowest concentration of amoxicillin/clavulanic acid (0.45% difference). In the case of three measures, the results differed by more than 10%—they were obtained for metronidazole, ciprofloxacin, and vancomycin at the concentration of 2 mg/L. For the *E. coli* strain, the greatest discrepancy in the results of both methods was obtained. In the case of 7 out of 10 measurements, a discrepancy significantly exceeding 5% was obtained. The use of vancomycin showed a difference of over 42% in the measurements, which was the highest discrepancy recorded in this study. In the time-dependent assays, significant discrepancies in the measurements were observed. In most cases, the values obtained during the CZE measurements were higher than those obtained on FC. In total, in 10 out of 15 measurements, the difference exceeded 10%. With increasing incubation time, a slight decrease in the difference in measurements between methods was observed for gram-negative strains and an increase for gram-positive strains.

### 3.5. Statistical Analysis

A Pearson’s test was performed to highlight the similarities between applied methods based on bacterial viability data. As variables, we considered the percentage of live cells obtained by FC and CE methods from the static study, the optical density and the peak area registered in the CE electrophoregrams. The Pearson’s test was chosen as a parametric statistical tool. It observed a linear correlation between the investigated variables and the data of bacterial viability. A correlation matrix was prepared for each bacterium separately (shown in Figure 7A) as well as for all bacterial strains together (shown Figure 7B). The correlation values significant at a *p* = 0.05 level ranged from 0.5008 to 0.9952, which pointed to a medium to very strong positive correlation. For all bacteria, a high correlation was observed between the percentage of viable cells obtained by the CF and CE techniques as well as between the OD value and peak area (>0.9), excluding the *P. aeruginosa* bacteria strain where the OD value was found to equal 0.87 (related to the lower optic density of thecontrol sample) (Figure 7A); however, this is very close to 1, suggesting a high correlation. In turn, *K. pneumoniae* strain was classified as a lowest correlated compared with rest investigated strains (round 0.50–0.53—slight correlation).

In the case of *E. coli*, statistically significant correlations were found between the viability in FC and OD (0.73), as well as between FC and the peak area (0.75). In the case of *P. aeruginosa*, these were the correlations between the peak area value and the viability determined in CE and CF, whereas for *S. aureus* and *E. faecalis*, all other correlations were statistically significant, and the range was 0.81–0.85 and 0.73–0.91, respectively. The summary correlation matrix including all strains also showed a very high (>0.95) correlation between viability in the FC and CE methods. In the case of OD and peak area the correlation was slightly lower (>0.84) but statistically significant. The slight correlation was noticed for the P, OD and CE and FC (around 0.6) (Figure 7B).

Additionally, a cut-off for the FC and OD methods has been established based on the obtained data regarding viability (Figure 7C). For this purpose, the obtained OD values were converted into percentages. Then, a scatterplot with box plots was generated to show the relationship between the mean values obtained during OD measurements and the corresponding bacterial viability effect obtained by FC. Based on the data obtained, the cut-off was found to be 50%. Therefore, in this context the cut off value will be considered only for viability and not for the inhibitory effect. Inhibitory effects of effective antibiotics decrease the total amounts of bacterial cells in comparison with the control (not-treated) bacterium.

## 4. Discussion

By affecting the metabolism or cellular structures of bacteria, antibiotics are able to inhibit their growth or kill them. The effectiveness of antibiotics on individual bacterial strains can be assessed by measuring their viability. This paper presents a method for the determination of live and dead bacterial cells using capillary zone electrophoresis and our own algorithms. To the best of our knowledge, this study is the first to successfully use CZE to quantify the viability of bacteria exposed to antibiotic stress.

Bacterial cells suspended in an electrolyte and exposed to an electric field show movement. It results from the fact that there are acid-base functional groups on the surface of cells which give them a surface charge and surface potential. The complexity of the structure of the cell wall determines its total surface charge. The surface charge of all living cells at physiological pH (between 5 and 7) is negative, because the number of carboxyl and phosphate groups exceeds the number of amino groups [17,18]. In gram-positive bacteria, the negative charge is caused by the presence of teichoic and teichuronic acid associated with peptidoglycan. In gram-negative bacteria, it is caused by the presence of lipopolysaccharides, phospholipids, and proteins on the surface of the cell membrane [19]. Living cells have systems (such as ion exchange and proton pumps) that contribute to the exchange of electrolytes and ensure homeostasis. In the case of dead cells, these processes are stopped, which changes the surface charge [20]. The potential of cells changes as their state changes. In an alkaline environment, the zeta potential of dead cells is less negative than the values determined for living cells [21]. In the present experiment, we have used two buffers with a pH above 7, which allowed for the deprotonation of functional groups of both the inner wall of the capillary and the surface of the bacteria [22]. The works of Buszewski et al. show that there is a dependence of deprotonated functional groups, the emerging zeta potential and its translation into the degree of dispersion of bacterial cells [15,17,21,23]. The high zeta potential keeps cells apart due to repulsive surface forces. Electrophoretic methods have some limitations due to difficulties in interpreting electropherogram, as cells tend to cluster. Aggregation and adhesion have always been a problem in the electrophoretic separation of microorganisms. In order to limit their influence on the analysis result, modified capillary fillings, various buffers and additives were used, or the EOF was changed so that it was possible to separate bacteria [24]. Dziubakiewicz et al. indicated that bacteria can be efficiently analysed by CZE after modification of the surface charge of microorganisms by calcium ions, which bind to their surface and significantly reduce the repulsive forces [25]. The surface modification affects the sharpening of peaks, reducing their number and improving the baseline shape [26]. In order to make aggregations independent of the influence of zeta potential, in this study, we have suspended the bacteria in calcium nitrate, which resulted in the production of a “controlled aggregate” (clumping). During the analysis, compact electropherograms were obtained and the detection of most of bacteria was possible in the form of a single compact band. This facilitated the determination of the retention time and calculation of the surface area of the signals coming from the bacterial cells, and therefore the weakness of the CZE method became its advantage. For the tested strains, a very strong correlation (the coefficient of correlation obtained by the Pearson’s test was 0.85) was obtained between the optical density values and the area of the electropherogram peak. The coefficient of determination was the highest for *E. coli* (0.9988) and the lowest for *S. aureus* (0.9922). For all strains, SD value for OD and peak area (CE) ranged from 0.00 to 0.05, and 5.63 to 59.36 respectively. Thus, the optical density corresponds to the area of the electropherogram peak, which transfers to the number of cells in the suspension. Crispo et al. also obtained a high correlation between these values for the three yeast species [27]. In another study, they showed that the shape and intensity of the peak appeared to be key in the electrophoretic discrimination of live and damaged *Oenococcus* subpopulations after ethanol exposure [27]. The work of Sautrey et al. confirms that CZE can also be used in studies of the effects of antibiotics on bacteria. Using this method, they showed that the surface charge, and thus the electropherogram of gram-negative bacteria treated with colistin was different in the case of sensitive and resistant strains. In this experiment, we used the migration time as a parameter differentiating the action of antibiotics on the viability of bacterial strains. These values were determined for both live cells incubated under non-stress conditions and cells inactivated by MeOH (dead-cell control). The analysis of bacterial cells showed that the dead cells had a shorter migration time to the detector than the live cells. This is consistent with changes in the values of surface charge and zeta potential described above. We found these values constant and, using them, we calculated the percentage of viable cells treated with different antibiotics.

The antibiotics used in this experiment showed different effects on the tested strains as they have different mechanisms of action. Reference strains were analyzed because their sensitivity to antibiotics (due to the lack of acquired resistance) is easy to predict and is based on both European and American guidelines. Amoxicillin and cefotaxime belong to beta-lactam antibiotics, which work by binding to proteins on the surface of the bacterial cell, which inhibits the cross-linking process in cell wall synthesis and leads to cell lysis. For this reason, these antibiotics are definitely more effective against gram-positive bacteria [28]. Antibiotics belonging to this group are often administered together with a beta-lactamase inhibitor (like clavulanic acid), which allows increasing the spectrum of their action with strains producing the beta-lactamase enzyme [29]. The bactericidal activity of vancomycin is also based on this mechanism [30]. Since the action of these antibiotics leads to the destruction of bacterial cells and, as a consequence, their fragmentation, electrophoretic measurements show greater dispersion stability for systems with dead cells, compared to living cells, which tend to aggregate. Ciprofloxacin prevents DNA replication by inhibiting the action of enzymes involved in this process and is mainly used to treat infections with gram-negative strains [31]. Clindamycin inhibits the synthesis of bacterial proteins by binding to the 50 S ribosomal subunits and is dedicated mainly to gram-positive bacteria [32]. In turn, the mechanism of action of metronidazole is related to the reduction of nitro groups and the formation of free radicals, which inhibit the synthesis of nucleic acids. It is used for both gram-positive and negative bacteria [33]. The selection of antibiotics and their concentrations for the tested strains was based on the EUCAST and CLSI guidelines. In this experiment, antibiotic influence on the viability of bacteria have been studied over a wide range of concentrations (antibiotic susceptibility assays) and time incubation (time-dependent assays). The applied antibiotics limited the viability of bacterial cells to a varying degree. Moreover, the mean number of viable cells decreased with the increase of the concentration of the antibiotic added to the test sample as well as with the lapse of the incubation time with the high concentration of the drug. The research results obtained in static experiment with the use of CZE showed that in the case of *E. faecalis*, the use of vancomycin allowed for the greatest reduction in cell viability. For *S. aureus*, the lower amoxicillin concentration and the higher ciprofloxacin concentration were the most effective. In the case of *Enterobacteriacae*, vancomycin and amoxicillin with clavulanic acid applied in high concentrations proved to be the most effective. In turn, on *P. aeruginosa* cells, the best effect was achieved by the use of high doses of cefotaxime. The results obtained with the CZE were compared with the results obtained with the reference method—flow cytometry. In all of the studies, the increasing concentration of the antibiotics was strongly correlated with data obtained from the CZE and CF measurements—The coefficient of correlation obtained by the Pearson’s test for CE and CF data was 0.95. In total, in 12 out of 50 measurements were performed, and the difference between the measurements was observed at RSD of approximately 10%. In the case of the time-dependent experiment, similar differences in measurements were obtained more often (10 out of 15 measurements). The percentage of viable cells obtained with both methods in some measurements was divergent, however, when describing their effectiveness with the IC50 index, their occurrence was not observed. The IC50 (inhibition concentration) is a quantitative measure of the toxicity of a test substance [34]. In this case, it determines the concentration of the antibiotic that reduces the viability of bacterial cells by half. Figure 8 presents the results obtained using the CZE and FC methods with the IC50 concentrations given in the recommendations. The results obtained in this way in most cases coincide with the recommendations, which proves the sensitivity of the reference strains selected for testing to the antibiotics used, and thus the possibility of using the CZE technique to determine cell viability. The red colour in the table shows the strains for which too low concentrations of antibiotics were used, on the basis of which it is not possible to unequivocally determine their effectiveness.

## 5. Conclusions

In summary, capillary zone electrophoresis appears to be effective in determining the viability of pathogens exposed to substances that change their physiological state. The use of an algorithm that takes into account the detection time is the basis for determining the ratio of live and dead cells in the tested sample. On the other hand, on the basis of the size of the electrophoretic signal, one can indirectly infer the number of bacterial cells in the tested sample, but not their viability. There is a relationship between the optical density measured spectrophotometrically and the surface area of the electropherogram, as evidenced by the calibration curves of all tested strains. There is a high agreement between capillary electrophoresis and flow cytometry in the assessment of the percentage of live cells exposed to the stress factor in both antibiotic susceptibility and time-dependent assays. The applied measuring system to assess the effectiveness of antibiotic therapy in in vitro conditions is a method with great potential, and the data obtained with the use of CZE mostly correspond to the expected drug sensitivity. Use of the IC50 criterion can serve as a benchmark in categorizing antibiotic susceptibility. Capillary zone electrophoresis may be a tool that enables conclusions about the drug susceptibility of clinical strains; however, its application requires further research.

## Figures and Tables

**Figure 1 ijerph-19-01833-f001:**
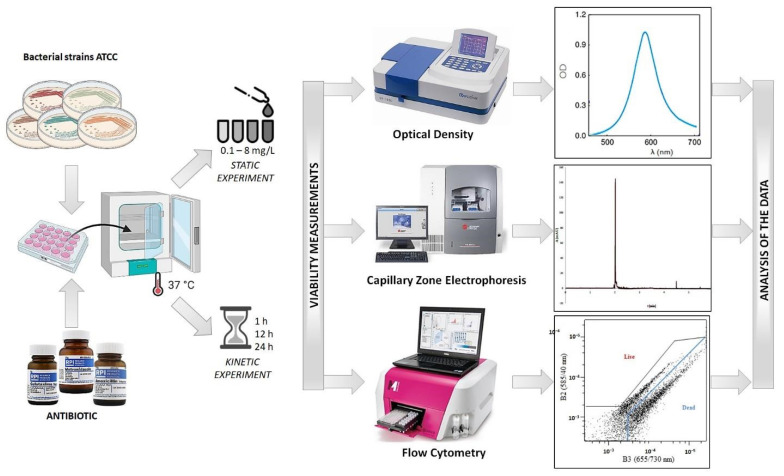
Scheme of workflow.

**Figure 2 ijerph-19-01833-f002:**
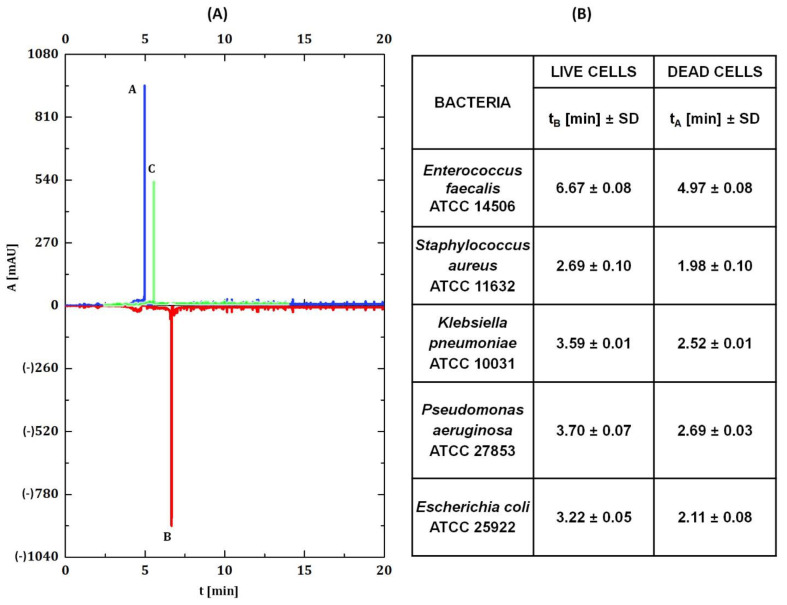
(**A**) Illustrative electropherogram obtained on the basis of the described calculations on the example of E. faecalis, A—dead cells (*t_A_*), B—live cells (*t_B_*), C—cells treated with an antibiotic (*t_i_*). (**B**) Table presents the mean retention time with the standard deviation (SD) of the tested bacterial strains.

**Figure 3 ijerph-19-01833-f003:**
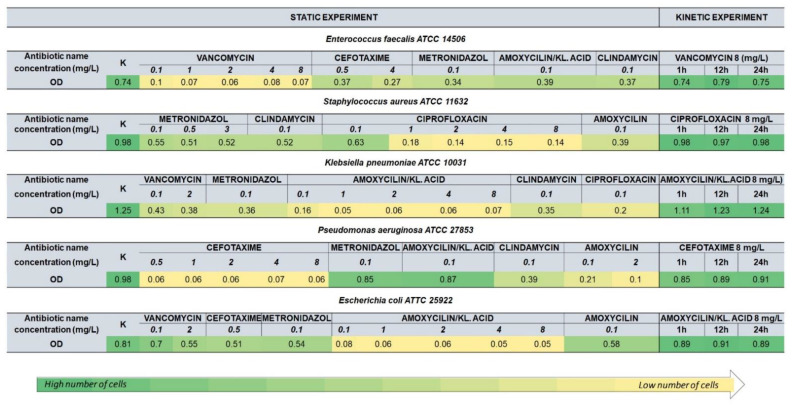
Summarized the value of bacterial cell viability under antibiotic treatment expressed as an OD value.

**Figure 4 ijerph-19-01833-f004:**
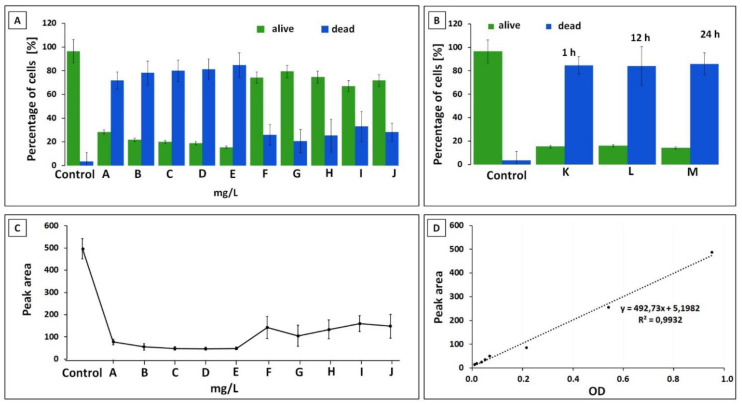
Percentage of live and dead bacterial cells calculated according to the proposed Equations (1) and (2) for the *E. faecalis* strain obtained using CZE in: (**A**) the static experiment, (**B**) the dynamic experiment (markings on the charts: A—vancomycin 0.1 mg/L, B—vancomycin 1 mg/L, C—vancomycin 2 mg/L, D—vancomycin 4 mg/L, E—vancomycin 8 mg/L, F—cefotaxime 0.5 mg/L, G—cefotaxime 4 mg/L, H—metronidazol 0.1 mg/L, I—amoxiciolin with clavulanic acid 0.1 mg/L, J—clindamycin 0.1 mg/L, K–M—vancomycin 8 mg/L). (**C**) The total area of signals obtained during the analysis of *E. faecalis* in the static experiment. (**D**) Dependence of absorbance (OD) on the area of the electropherogram peaks for the Enterococcus faecalis. The graph shows the trendline equation and the coefficient of determination. The data presented in the Figure 4D regarding the correlation between OD and peak area have been reported as a results of three repetitions of each with the SD value from 0.01 to 0.02 (OD) and from 6.28 to 49.36 (Peak area).

**Figure 5 ijerph-19-01833-f005:**
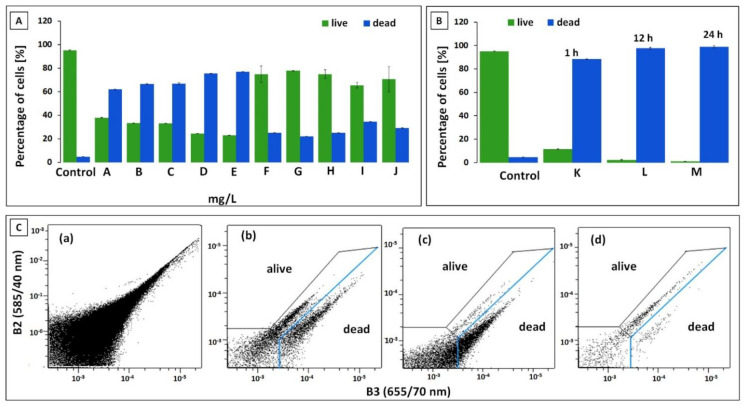
Percentage of live and dead bacterial cells for the *E. faecalis* strain obtained using FC in: (**A**) the static experiment, (**B**) the dynamic experiment (markings on the charts: A—vancomycin 0.1 mg/L, B—vancomycin 1 mg/L, C—vancomycin 2 mg/L, D—vancomycin 4 mg/L, E—vancomycin 8 mg/L, F—cefotaxime 0.5 mg/L, G—cefotaxime 4 mg/L, H—metronidazol 0.1 mg/L, I—amoxiciolin with clavulanic acid 0.1 mg/L, J—clindamycin 0.1 mg/L, K–M—vancomycin 8 mg/L). (**C**) Histograms obtained from FC analyses: (**a**) *E. coli* control sample, (**b**) *E. coli* incubated with amoxicillin at the concentration of 1 mg/L, (**c**) *E. faecalis* incubated with vancomycin at the concentration of 1 mg/L, (**d**) *P. aeruginosa* incubated with cefotaxime at a concentration of 1 mg/L.

**Figure 6 ijerph-19-01833-f006:**
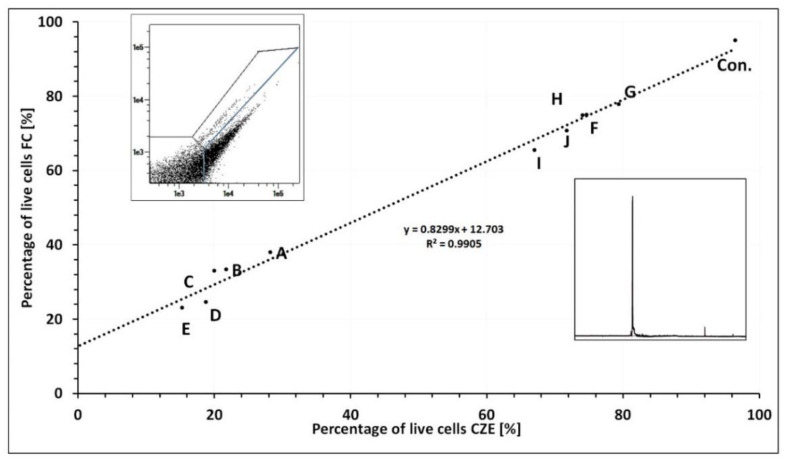
Comparison of *E. faecalis* viability measurements made with two different methods, i.e., FC—flow cytometry and CZE—capillary zone electrophoresis. The graph shows the trendline equation and the coefficient of determination (R2). The labels of the punks refer to the antibiotic used (Con.—control without antibiotic, A—vancomycin 0.1 mg/L, B—vancomycin 1 mg/L, C—vancomycin 2 mg/L, D—vancomycin 4 mg/L, E—vancomycin 8 mg/L, F—cefotaxime 0.5 mg/L, G—cefotaxime 4 mg/L, H—metronidazol 0.1 mg/L, I—amoxiciolin with clavulanic acid 0.1 mg/L, J—clindamycin 0.1 mg/L). The data presented in this figure regarding the correlation between CZE and FC have been reported as the results of three repetitions of each with the SD value from 4.62 to 20.05 (CZE) and from 0.09 to 10.74 (FC).

**Figure 7 ijerph-19-01833-f007:**
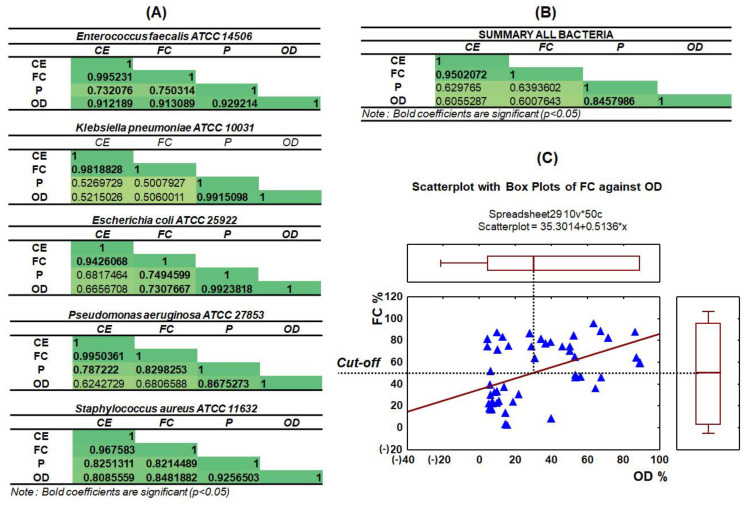
Correlation matrix showing the significance between FC, CE, the optical density method and the peak area value for each bacterium separately (**A**) and the summary aspect of all used methods for all investigated bacterial strains (**B**). Cut-off value generated by the Scatterplot and box plots based on OD and FC data (%) (**C**). * CE = CZE.

**Figure 8 ijerph-19-01833-f008:**
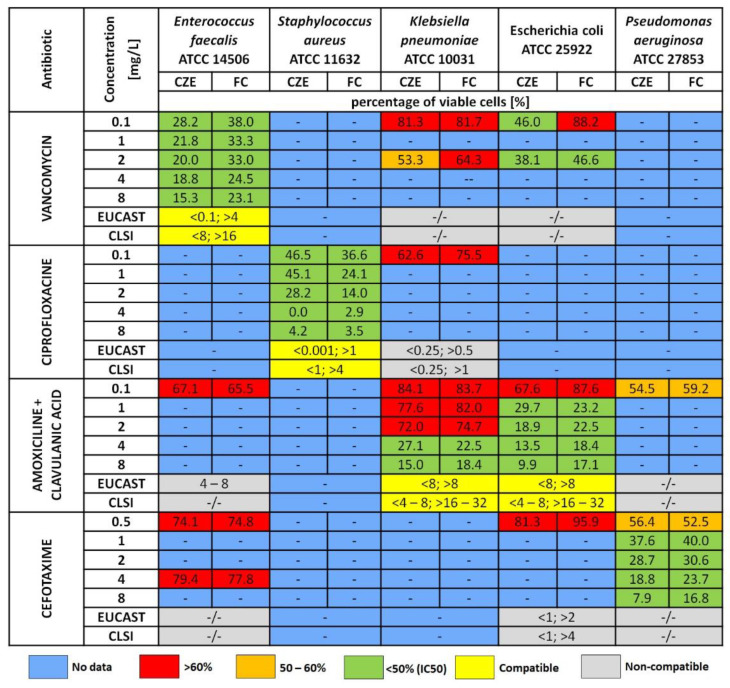
Summary of data obtained using the CZE and FC method with reference to the EUCAST recommendation, and comparative CLSI. According to both recommendations, the IC50 is shown (the value < x represents the concentration below which the strains are considered susceptible to the antibiotic; the value > x represents the concentration below which the strains are considered resistant to the antibiotic). The colours are shown: blue—viability < 50% (IC50) determined, yellow—viability 50–60%, orange—viability > 60%, green—the obtained results are in line with the recommendations, red—the obtained results are not in line with the recommendations, grey (−/−)—no data.

## Data Availability

Not applicable.

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
