# Peer review of "Capillary Zone Electrophoresis in Tandem with Flow Cytometry in Viability Study of Various ATCC Bacterial Strains under Antibiotic Treatment"

_ijerph, 2022, doi:10.3390/ijerph19031833_

Round 1
Reviewer 1 Report
there are no description of correlation coefficient nor in the text neither in the caption of figures, instead in statistical methods chapter authors presented correlation as a method to evaluate relation. The result of the t-test to evaluate significance of correlation is not reported in the mai text.
The description of statistical analysis include correlation and GLM to determine regression line, but its not stated wich experiment is analysed by GLM and why GLM and not simple linear regression. Are a link function necessary to model data?
R-square is a measure of the fitting of the model, it is not appropriate to use the coefficient of determination as measure of correlation. The authors should choose to express results as highly correlated, showing Pearson or Spearmann correlation coefficient or as line with b=1 (that means perfect correspondence between x and y) with an high fit of data (that is R-square).
In figure 4 d if main aim wasn't to study the prediction of Peak by OD, but only relation between Peak and OD, the equation is unuseful. THe Pearson COrrelation Coefficient is more appropriate.
In the Statistical analysis the authors stated the t-test for correlation, but there's no p-values. Of course R-square has no p-values, therefore it's better to choose what results to show.
in discussion line 467 (and following comments on correlation) should be revised accordingly to previous comment.
line 470: insignificant it's not a nice term please describe better the inconsistency of the error
line 522 and following, please revise in the light of previous comment.
Author Response
Answear to the Reviewer Comments and Changes Made
We would like to thank the Reviewers for careful reading, and constructive suggestions for our manuscript that will help us to improve our work. According to the comments from the reviewers, we comprehensively revised our manuscript. Hoping that we addressed all the questions mentioned by the reviewers, below we include the point-to-point response to each comment.
In the manuscript file all the changes have been provided with using the “Track Changes” function.
Reviewer(s)' Comments to the Author:
Reviewer: 1
Comment: There are no description of correlation coefficient nor in the text neither in the caption of figures, instead in statistical methods chapter authors presented correlation as a method to evaluate relation. The result of the t-test to evaluate significance of correlation is not reported in the main text.
The description of statistical analysis include correlation and GLM to determine regression line, but its not stated wich experiment is analysed by GLM and why GLM and not simple linear regression. Are a link function necessary to model data?
R-square is a measure of the fitting of the model, it is not appropriate to use the coefficient of determination as measure of correlation. The authors should choose to express results as highly correlated, showing Pearson or Spearmann correlation coefficient or as line with b=1 (that means perfect correspondence between x and y) with an high fit of data (that is R-square).
In figure 4 d if main aim wasn't to study the prediction of Peak by OD, but only relation between Peak and OD, the equation is unuseful. THe Pearson COrrelation Coefficient is more appropriate.
In the Statistical analysis the authors stated the t-test for correlation, but there's no p-values. Of course R-square has no p-values, therefore it's better to choose what results to show.
line 470: insignificant it's not a nice term please describe better the inconsistency of the error.
line 522 and following, please revise in the light of previous comment.
Answer: We would like to thank the Reviewer for the important comments. We apologies for such inconvenience. The “GLM” abbreviation has been used by mistake, therefore, the description of used statistic methods was revised and improved. Moreover, the Reviewer’s suggestions have been taken fully in consideration and additional statistics assays have been performed. All the changes are included in the manuscript.
Comment: in discussion line 467 (and following comments on correlation) should be revised accordingly to previous comment.
line 470: insignificant it's not a nice term please describe better the inconsistency of the error
line 522 and following, please revise in the light of previous comment.
Answer: Thanks the Reviewer for the important comment. After conducting additional statistical analyzes, we changed the wording of the above-mentioned lines.
Reviewer 2 Report
The present work developed a method to examine bacterial survival using the capillary zone electrophoresis (CZE) and flow cytometry (FC). The antibiotic resistance of different strains of bacteria was compared using these two methods. The article is well written and easy to understand.
Some principal points should reviewed to improve the article and make it more compressible:
- mention in the introduction the guidelines on antimicrobial susceptibility testing ;
- In figure 3 make the OD values visible ;
- establish a cut-off for FC method to determine whether or not there is bacterial growth in relation to the OD value;
- In sections 3.1, 3.2 and 3.3 many of the data shown do not correspond to the figures and/or supplementary data. In the same sense, figure 3 is not cited in paragraph 3.1.
Author Response
Answear to the Reviewer Comments and Changes Made
We would like to thank the Reviewers for careful reading, and constructive suggestions for our manuscript that will help us to improve our work. According to the comments from the reviewers, we comprehensively revised our manuscript. Hoping that we addressed all the questions mentioned by the reviewers, below we include the point-to-point response to each comment.
In the manuscript file all the changes have been provided with using the “Track Changes” function.
Reviewer 2:
The present work developed a method to examine bacterial survival using the capillary zone electrophoresis (CZE) and flow cytometry (FC). The antibiotic resistance of different strains of bacteria was compared using these two methods. The article is well written and easy to understand. Some principal points should reviewed to improve the article and make it more compressible:
Comment: mention in the introduction the guidelines on antimicrobial susceptibility testing;
Answer: Thanks for the Reviewer valuable remark. We added this information to the Introduction.
Comment: In figure 3 make the OD values visible;
Answer: Thanks the Reviewer for the important comment. We have improved the visibility of values. The new changed Figure has been added to the manuscript.
Comment: establish a cut-off for FC method to determine whether or not there is bacterial growth in relation to the OD value;
Answer: Thanks the Reviewer for the important comment. We generated a scatterplot with box plots to show the relationship between the mean values obtained during OD measurements and the corresponding bacterial viability effect obtained by FC. Based on it we establish a cut-off for FC method. More information and the figure we presented in the manuscript.
Comment: in sections 3.1, 3.2 and 3.3 many of the data shown do not correspond to the figures and/or supplementary data. In the same sense, figure 3 is not cited in paragraph 3.1.
Answer: Thanks for the Reviewer valuable remark. We added to the manuscript text an information that the figures in sections 3.2, and 3.3 show are an example relating only to the strain E. faecalis. We also corrected the information on the data contained in the supplementary data, and added omitted figures quotes in the text.